# The Impacts of COVID-19 Pandemic on the Food Sector and on Supermarket Employees in France during the First Lockdown Period

**DOI:** 10.3390/healthcare10081404

**Published:** 2022-07-27

**Authors:** Cyrielle Dumont, Génia Babykina

**Affiliations:** ULR 2694-METRICS—Évaluation des Technologies de Santé et des Pratiques Médicales, CHU Lille, University of Lille, F-59000 Lille, France; evgeniya.babykina@univ-lille.fr

**Keywords:** COVID-19 pandemic, physical and physiological impacts, statistical analysis, supermarket staff, working conditions

## Abstract

During the first lockdown period due to the COVID-19 pandemic, from the 17 March 2020 to the 11 May 2020 in France, essential professionals (nursing staff, police officers, supermarket staff, etc.) continued to be physically present at their workplaces. The present study focuses on exploring impacts of the pandemic on supermarket staff and on the food sector in France: COVID transmission among supermarket workers, working conditions, food supply, etc. For that, two anonymous surveys were addressed to supermarket employees and to supermarket supervisors. In total, 1746 responses from employees and 171 responses from supervisors were recorded all over France. Over 70% of employees and almost 50% of supervisors were women and over 50% of employees were between 25 and 40 years old. The following main trends in terms of physical and psychological impacts are revealed: 7% of employees working during the lockdown reported having COVID, although a still poorly developed screening and lack of diagnostic tests during the first lockdown should be kept in mind. The working conditions changed; higher work load, a more stressful environment, inappropriate client attitude, a lack of recognition, fatigue, and shortages were reported. A lack of government recognition, namely no prime allocations to supermarket staff during the lockdown period, is also often mentioned. Finally, no priority was given for store employees in terms of childcare.

## 1. Introduction

The COVID-19 pandemic originated in China in December 2019, and then rapidly spread throughout the world. The outbreak in France began in February 2020. Due to the increasing number of cases and of patients admitted in hospitals, the French government decided to establish a lockdown from the 17 of March 2020 to the 11 of May 2020. According to the data provided by John Hopkins University (accessed on 12 July 2022, https://ourworldindata.org/coronavirus/country/france), the 7-day rolling average for the number of new confirmed cases in France was 847 and 1088, respectively, in the beginning (17 March 2020) and in the end (11 May 2020) of our study. The respective numbers of this rolling average for deaths was 17 and 206. During this period, essential professionals continued to be physically present at their workplace. It concerns, for example, nursing staff, police officers, or supermarket employees. Many studies which focused on treatment protocols and on virus transmission mechanisms, as well as on social phenomena, were carried out since the beginning of the pandemic. Among the latter, many papers addressed working conditions and mental state of healthcare workers [1,2,3,4,5,6], including comprehensive systematic reviews [7]. These studies underline a high risk of infection through direct workplace exposure and an increased work load and psychological stress. The consequences on mental health (mental injury) are mentioned to be important. Some longitudinal studies assessing the psychological impacts of COVID-19 pandemic do not show a significant increase in depression, anxiety, and stress [8,9,10,11], even though some particularities in prevalence are mentioned. A decrease in post-traumatic symptoms is reported in some studies [9,10], as well as an increase in general psychological distress [12,13]. Fewer studies concern the changes in consumption behaviour (refer to [14] for a study of food consumption patterns during the lockdown in Spain and to [15] for a study concerning the dynamics of supermarket clients’ behaviour change in Germany). The working conditions of supermarket employees and their mental state, however, lack attention. Still, these are essential front-line employees mobilised to maintain food supply during the lockdown. To our knowledge, one study concerned the virus transmission among this population in China; more specifically, it approached a cluster epidemic caused by one imported case of COVID-19 in a supermarket of Liaocheng city (China) [16]. In a Ying et al. report [17], the authors approach the specific modelling of COVID-19 transmission in supermarkets using an agent-based model. General issues concerning the employees’ conditions, not specific to the food sector or to COVID-19, were approached, for example, in the Roach study [18] (factors of job satisfaction), in the Saunderson survey [19] (general employee recognition), or in the Boye et al. report [20] (organisational culture and productivity). The papers rarely focus on supermarket employees. A general literature review on the essential workers’ conditions (not specific to supermarket employees) during the pandemic is provided in Gaitens et al. review [21]. In the Dennerlein et al. report [22], the authors propose recommendations for working conditions based on guidelines. To our knowledge, there is no empirical study targeting supermarket employees and supervisors and simultaneously approaching several issues. However, the high importance of addressing this population is emphasised, for example, in a recent comprehensive review [21]. In this context, in the present paper, we propose an exploration of working conditions, employee morale, sick leaves, turnover, and shortages by means of an exploratory pilot survey addressed to food retail workers during the first lockdown in France. The world may have to expect to deal with pandemic situations for a long time; thus, exploring the impacts of COVID-19 pandemic on essential sectors, such as food retailing, seems to be important. This pilot study can be used as a background for further targeted investigations with increased precision in the area. Moreover, the information outlined in the paper can be used as background for responses of food retail area managers to other widespread outbreaks and can allow them to anticipate problems caused by these outbreaks. Finally, the findings of the study can be taken into account by the authorities when developing social, sanitary, quality, and security regulations and measures.

The objective of the present pilot study is to explore the impacts of the pandemic on supermarket staff and on the food retail sector in general in France. To achieve this objective, two surveys were carried out: one addressed to supermarket employees and the other to supermarket supervisors. These surveys aimed to explore general trends in working conditions and COVID transmission among supermarket workers, as well as to obtain feedback concerning the employees’ mental and physical state during the lockdown period. Geographical trends, as well as changes in consumption, food supply, and turnover, were also investigated.

In this exploratory study, the following points are approached:A description of the sample of supermarket employees and supervisors in terms of socio-demographic characteristics, working conditions during the lockdown, COVID status, etc.A description of employees’ working conditions reported by supervisors.A description of leaves during the lockdown period (duration, reasons).An exploration of trends in terms of employees’ positions. In this sense, the following research questions are addressed: Do working conditions, risk of contamination, and moral impact change according to employee’s position? Do some employees have more difficulties than others?A description of financial impacts of lockdown (changes in turnover, orders).An exploration of trends in terms of shortage during the lockdown.An exploration of geographical trends in terms of COVID prevalence.Employees’ and supervisors’ morale, i.e., their general attitude, satisfaction, and overall condition.

## 2. Materials and Methods

### 2.1. Data Collection

To assess the general conditions and the changes in food supply stores during the COVID pandemic and to address the defined research questions, two anonymous independent surveys were developed: one addressed to supermarket employees (sales assistants, shelf managers, etc.) and the other addressed to supermarket supervisors; two different links were thus provided. The responses to both questionnaires were anonymously recorded. Data collection using a blind survey implies that the present study does not require an ethics statement. So, no ethics committee approval was necessary. Both questionnaires were created by means of Google Forms and were transmitted principally via the social network (the group “Je Bosse en Grande Distribution” in Facebook, including 239,057 members). The responses were collected during 1 month, from 24 June 2020 to 22 July 2020. The questionnaires were principally transmitted via social networks, and the respondents were informed about the study and voluntarily chosen to participate as a part of the sample group; the responses were obtained on a voluntary basis. Thus, no participant written consent was necessary.

The questionnaires included single-choice and open questions, leaving a large liberty of expression to respondents. No question was mandatory.

For this internet survey, a non-probability convenience sampling procedure was employed, i.e., no random selection of respondents was performed, and the targeted population was easy to reach [23,24]. Precisely, the members of the “Facebook” group, working as supervisors or employees in French supermarkets, responded to the survey in an anonymous way, and all responses were considered in the data analysis.

### 2.2. Statistical Analysis

Quantitative statistical methods were employed to analyse the obtained data and to address the research questions. Categorical variables were described by percentages with confidence intervals. Continuous variables were discretised and analysed as categorical variables. Discretisation was carried out in a manner to optimise interpretations. The chi-squared test or Fisher’s test were carried out to work out the independence between two categorical variables. Small *p*-values (<0.05) indicate a significant link between variables. Multiple factor analyses were employed to explore multivariate patterns in data. The multiple factor analysis is a dimension reduction method, allowing multi-dimensional categorical data to be illustrated in a simple graphical way. Using this approach, one can interpret graphical proximities between the values of categorical variables as correlations between these values. Correspondence analysis was used to visualise correlations between two categorical variables. Correspondence analysis is a dimension reduction method, allowing graphical proximities to be to visualised and interpreted between the values of categorical variables as links between these values. Textual responses were analysed by means of bar charts and word clouds. A word cloud is a graphical tool which can visualise textual data and extract general trends by applying a police size proportionally to frequencies of words employed in a text (frequently used words appear bigger than rarely used words). The data were analysed by R freeware, Version 1.2.5.

## 3. Results

The results of the survey analysis, providing responses to the defined research questions, are presented in the following order. Firstly, the characteristics of employees (socio-demographic status, COVID status, working conditions) during the lockdown are investigated. The employees’ morale is explored using the narrative textual approach based on their free comments. Multivariate trends are graphically explored.

Secondly, the responses of supervisors (working conditions of their employees and financial issues: turnover, orders, e-commerce) are investigated. Shortage and morale issues are analysed using the narrative textual approach based on their free comments.

### 3.1. Employees’ Responses Analysis

In general, the responses of 1746 employees were recorded all over France. The descriptive statistics are provided in Appendix A (Table A1). No specific trends are reported concerning the size of the store (various store sizes are represented). The majority of respondents were women (over 70%) and more than a half were 25–40 years old, and about 2/3 of respondents were shelf and self-service employees or checkout and reception assistants. Very few employees started the work in the store just before the lockdown (the current experience is more than 1 year for almost 90% of respondents). Over 95% of employees continued to work during the lockdown period, with a higher work load for over 80%. Finally, 30% of employees struggled to obtain the protection kit during the lockdown period.

Some questions on the personal information were not answered by the majority of respondents; it concerned information about the household size and questions covering COVID, if a person had COVID, if a person was tested for COVID, and if there were COVID cases in his/her household. For the three latter variables, no information is available on missing values (if missing values correspond to disease-free or to response refusals).

As for work and working conditions during the lockdown period, a detailed description of people working and absent during the lockdown is provided in Appendix A (Table A2, *p*-values of the chi-square test are also given).

In sum, 1671 of employees (95.7%) worked during the lockdown, 60 reported not working, and 15 responses were missing. Globally, the distribution of different characteristics of those who worked and did not work during the lockdown did not differ considerably. Slight trends are present for gender (men are more likely to work, chi-squared *p*-value 0.092) and for age (younger people are more likely to work, chi-squared *p*-value 0.069). Some significant but partly mechanistic correlations are also present: there are more employees for whom the work amount tended to increase and who struggled to obtain the protection kit among those who worked during lockdown, and, on the other hand, there were slightly more people with COVID among those who did not work.

A significant trend is also revealed for household size and for symptoms in households; however, this result should be interpreted with caution due to a large amount of missing responses.

The reasons of not working (*n* = 60, one person did not respond to the question) are quite equally distributed (refer to Table 1). Slightly more than a third did not work for child care reasons (note that schools were closed during the whole lockdown period), the same proportion did not work for sick leave reasons, and those remaining did not specify a reason.

Among those who worked, 194 (11.6% from the total of 1671) reported to be sick. A detailed description of this population is provided in Table 2. It mainly concerns women (75.3%), not recently employed (less than 12% employed since less than 1 year), working in hyper- or supermarkets (less than 50% are grocery store workers), mainly (almost 50%) shelf employees and checkout/reception managers. Note that 40.6% of sick employees experienced difficulties in obtaining the protection kit (vs. 30.4% among non-sick employees, *p*-value = 0.005). Among sick employees, there were significantly more checkout/reception managers (20% vs. 13% for non-sick employees) and less shelf employees (29% vs. 39% for non-sick employees) (*p*-value = 0.02). The duration of sick leave did not exceed 1 week for over 50%. No other particular trends were revealed.

Among those who worked during the lockdown, 113 reported to have COVID, which corresponds to 7% among those who worked and 61% among those who answered. Over 85% of those who worked reported an increased amount of charge (refer to Table 3 for detailed numbers).

Finally, 109 employees reported a decrease in work amount during the lockdown, opposing the general trend. These employees are characterised by the following features: women (60%) working in hypermarket (61%), and self-service or shelf employees (over 60%). One can refer to Figure 1 for details on positions of employees who reported a decrease in work load.

Concerning the uniform description, 913 employees (53%) reported no change in uniform. For 92% (n=754) of those for whom uniform changed, the change was initiated by their management, and others changed uniform on their own. The changes principally meant obtaining masks, gloves, and protection screens (41%, 17%, and 12% of responses, respectively), compared to 6% of responses indicating wearing masks and/or gloves in normal time. Furthermore, 31% of employees struggled to obtain a protection kit; this difficulty especially concerned self-service employees, administrative staff, and sales assistants.

As for the multivariate trends, one can refer to the results of the multiple correspondence analysis, illustrated in Figure 2. The analysis is focused on the following variables: being sick (yes/no), struggling to obtain a protection kit (yes/no), the position of the employee, and changed mood during the lockdown (yes/no). In total, 1519 respondents with no missing values for the considered variables were included in this descriptive analysis. The presented first factorial plan summarises over 20% of total information in the initial data. No clear trends are observed. However, the difficulty of obtaining a protection kit seems to be linked to the position (self-service employees, administrative staff, and sales assistants are more concerned by the difficulty). Check-out and reception employees tend to be sick more often than others and staff not sick during the lockdown are more likely to be subject to changes in mood.

The employees were asked for a free description of their mood change reason during the lockdown as well as for a general comment on the situation. The corresponding word cloud provided in Figure 3 reveals stress, fatigue, clients’ attitude, and anxiety as main factors influencing the mood change. More details are provided in free comments, summarised in Figure 4. The free comments were analysed textually and categorised into the following general groups: a lack of consideration (a lack of child care help, no prime, a lack of recognition by clients/supervisors); a lack of protection (no COVID-19 test, difficulty to obtain protection kit (gloves, masks, etc.)); clients’ attitude (aggressive attitude, rudeness, non-adherence to social distancing rules during the COVID-19 lockdown, etc.); stress/fatigue (stress, fatigue, back pain, etc.); work load (an increase in work load); and recognition (positive comments concerning the primes, possibility to obtain protection kit, recognition by clients and supervisors).

No particular geographical trends were observed in responses given by the employees to different questions. The result (namely, the first factorial plan) of the correspondence analysis, carried out to explore a link between the sickness and geographical region, is provided in Figure 5. In general, there were more COVID cases in Ile-de-France and Provence-Alpes-côte d’Azur; this trend corresponds to the general pandemic propagation patterns for Ile-de-France, but the link is less clear for Provence-Alpes-côte d’Azur during this period. Foreign employees (Belgium, Luxembourg) tended to report not being sick more frequently; however, these results should be interpreted with caution due to a very small number of foreign employees (the study concerned mainly French stores, only 29 employees were from bordering countries).

### 3.2. Supervisors’ Responses Analysis

In general, the responses of 171 supervisors were collected. The descriptive statistics are provided in Appendix A (Table A3). The respondents were equally distributed in gender and were mainly employed since more than 1 year in a store (less than 20% were employed for less than 1 year). Over 56% of responses concerned supermarket supervisors (vs. hypermarkets and grocery stores). In terms of the number of employees, as well as the weekly number of clients and turnover, the distribution appears to be quite homogeneous. E-commerce was present in almost half of the stores during normal time (45.6%), with no particular trends regarding the amount of e-commerce (turnover, number of clients).

The details on working conditions reported by supervisors are provided in Table 4. The results reveal 12% of absences during the lockdown period on average, 4% of sick leaves for COVID, and 6% of leaves for child care (the percentage is calculated from the total number of employees). Moreover, 58.5% of supervisors reported a uniform change during the lockdown for their employees, with 42% omitting to indicate who initiated the uniform change; 43.6% of supervisors struggled to obtain protection kits for their employees. Note that the majority of supervisors (92%) reported a change in work amount during the lockdown. Among those who reported changes in work amount, the majority (92%) of respondents indicated an increased work load.

In terms of turnover and number of orders, in general, lockdown implied changes in the number of clients, in turnover, and in the work amount for about 90% of stores. The details of these changes during the lockdown are provided in Table 5. Significant changes were observed during the lockdown compared to normal time as for the number of visits, the number of e-commerce orders, the global turnover, and the e-commerce turnover. Precisely, during lockdown, the number of clients, the number of e-commerce orders, and the corresponding turnovers tend to increase (refer to Appendix A and Figure A1 for illustration). Most of responses concerning the e-commerce turnover and quantity were relevant. Note however that one respondent reported had no e-commerce during the normal period, but had [30, 200) orders on-line weekly. Symmetrically, one respondent reported having e-commerce, but 0 weekly orders. The results are presented for actual responses. There were no such irrelevances in the reported e-commerce turnover. Most supermarkets (91%) did not modify their practice in terms of e-commerce (those who practiced or continued to practice, as well as those who did not, have not started e-commerce activity). There were very few changes: three respondents reported opening the e-commerce activity during lockdown and six respondents ceased the on-line activity during the lockdown. Half of the supervisors (n=84) reported an increase in the number of online orders and almost half (n=82) reported an increase in the e-commerce turnover during the period. One supervisor reported an unchanged number of on-line orders. The remainder of respondents did not reply to these two questions. Most of the supervisors (73%) reported an increase in the general turnover, 15% reported a decrease in the turnover, and 12% did not respond.

Among supervisors, 67 (39%) reported no change in uniform. For the remainder, the changes principally involved masks, gloves, and protection screens (47%, 29%, and 25% of responses, respectively), while 5% of responses indicated wearing a mask and/or gloves in normal time.

Over 95% of stores suffered from different types of shortages during the lockdown (two supervisors did not report shortage and five omitted to reply to the question). Higher demand and, to a considerably lower extent, a lack of supply appeared to be the main reasons of shortage (refer to Figure 6 for shortage reasons). Shortage principally concerned essential products (baking ingredients, cheese), cleaning and antibacterial products, and toilet paper. Note that there was also a particular shortage of printing ink, caused by distance working and schooling during the lockdown. The word cloud derived from the supervisors’ free responses concerning shortage is presented in Figure 7.

In addition, 35% of supervisors experienced mood change during the lockdown. The description of this change, provided by supervisors in their free comments, is summarised in Figure 8. The supervisors are principally concerned with stress and fatigue, clients’ aggressive and/or disrespectful attitude, and work load. Note that mood change reasons of supervisors are not quite similar to those reported by the employees, and that employees are more affected by a lack of recognition than supervisors, whereas stress and fatigue were considerably predominant reasons for mood change among supervisors.

### 3.3. Summary of Results

Most of the respondents worked in the corresponding stores for more than one year in the beginning of lockdown (over 90% of employees and over 80% of supervisors). Globally, few people were absent during lockdown: over 95% of employees continued to work and 12% of employees’ absences were reported by the supervisors. The main reasons for absences were sick leave and/or child care (note that schools were closed during this first lockdown in France). Moreover, 30% of employees struggled to obtain protection kits and almost 44% of supervisors mentioned struggling to obtain protection kits for their employees. Over 50% of employees and 40% of supervisors mentioned no uniform change during the lockdown. For the employees, the uniform change was mainly initiated by their supervisors (92%) and principally concerned masks, gloves, and protection screens. Both employees and supervisors reported a higher work load during the lockdown. In particular, the number of e-commerce orders and the e-commerce turnover increased. Shortages for some products were mentioned by 164 supervisors among 171. This shortage was implied by a higher demand (both traditional and e-commerce) as well as by a lack of supply. This shortage mainly concerned essential products, cleaning and antibacterial products, and toilet paper.

The general situation implied mood changes for both employees and supervisors. Employees were mainly affected by clients’ inappropriate attitude and by a lack of recognition from their supervisors or from clients, whereas supervisors were mainly impacted by stress and fatigue.

## 4. Discussion

The present study is focused on the situation of food stores during the COVID-19 lockdown in France in 2020. In total, 1746 questionnaires were collected from employees and 171 questionnaires were collected from store supervisors. In the context of the general situation in France during the studied period, the results can be interpreted as follows. Employees reported suffering from a lack of recognition by the government, by clients, and by their supervisors. Indeed, no prime was allocated by the government to supermarket staff at this time (even if the prime was provided by the store supervisors in some cases). Finally, store employees were not given priority for child care facilities. Employees underline these elements in free comments: compared to medical staff who faced an increased work load and poor working conditions but had priority for childcare accommodations and were highly supported by the population, the food store employees experienced increased work load, difficulties in childminding facilities, aggressive attitudes, and a lack of recognition. To summarise, this study shows that the supermarket staff had to work in stressful conditions: increased work load, inappropriate clients’ attitude, and excess of fatigue. These conclusions join the points outlined in [21], a recent narrative study on essential workers in the United States, mentioning an increased risk of moral injury due to specific work-related factors. Moreover, as the respondents of our study, the authors of this paper conclude: “While essential workers in the health sector are hailed as heroes, other essential workers are treated as expendable”. This feeling seems to be shared by many essential workers rather than health employees. Our study does not underline a lack of personal protective equipment due to employer’s failure, or an increased work-related illness, as is the case in [21]. It is important to mention that labor legislation and general working conditions are rather different in France and in the United States; thus, comparisons should be carried out with caution.

In terms of contamination, the obtained results should be interpreted with care. Indeed, COVID screening was poorly developed during the first lockdown due to a lack of diagnostic tests, leading to underestimated prevalence.

A recently published study [17] assesses the effectiveness of different policies in terms of virus transmission reduction, such as restricting the number of customers in a store or bringing changes to the store layout. The authors propose an agent-based model of customer movement in a supermarket with a simple viral transmission model, based on the amount of time a customer spends in close proximity to other infectious customers. This model will potentially allow the retailers to propose effective policies to reduce virus transmission and, thereby, to protect both customers and staff. This new investigation underlines the importance of the topic treated in our study.

### Study Limitations and Further Work

The present study is principally focused on psychological and behavioural issues; the points concerning pandemic and virus propagation were approached subjectively using an auto-reported questionnaire. It would be helpful to enrich our results using a more objective study focused on contamination process during shopping. A more recent wide-spread diagnostic testing procedure can optimise the hygiene and safety guidelines and reduce disease propagation. Note that similar ongoing, but not yet published, studies exist for concert halls.

In terms of survey methods, an internet-based survey with non-probability sampling implies an important selection bias. Moreover, our study lacks a clear definition of the target population. Thus, only general conclusions can be drawn from it. We emphasise that the objective of our exploratory pilot study involves capturing the main trends which are important or interesting to develop in further investigations. Indeed, no other quantitative studies have been conducted on this topic to our knowledge. More targeted studies should be carried out, focusing on one main research question to clearly define a target population, to provide survey accuracy and to assess sample representativeness (for example, a study on moral injury, a study on contamination, a study on sick leaves, a study on turnover/orders issues, etc.).

## 5. Conclusions

Our pilot study addresses the impacts of the COVID-19 pandemic on the food sector and on working conditions of supermarket employees in France, revealing a lack of financial and social support of supermarket employees in the context of pandemic. In particular, no prime allocation and no aid in terms of childcare facilities was provided by the French government. However, supermarket employees are essential “front-line” workers in this situation and are subject to higher risks for their physical and mental state; this population should be closely considered in the case of acute epidemics.

## Figures and Tables

**Figure 1 healthcare-10-01404-f001:**
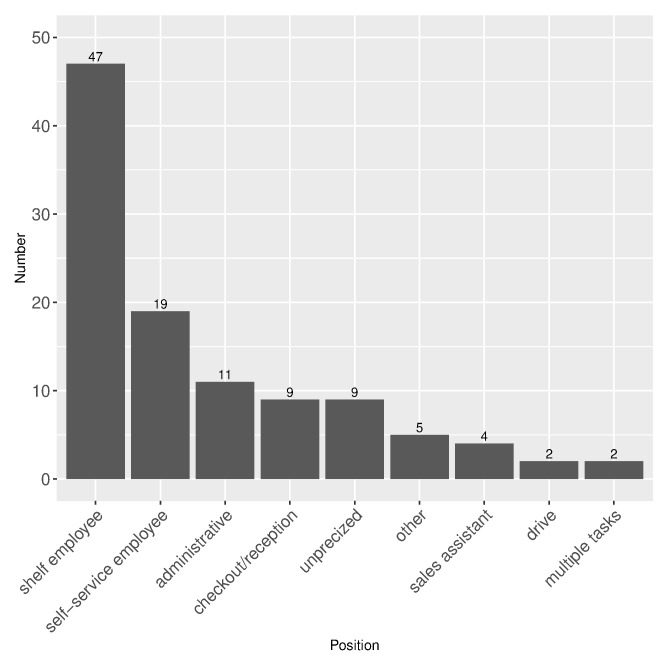
Positions of employees who reported a decrease in work load (n=109, one missing answer).

**Figure 2 healthcare-10-01404-f002:**
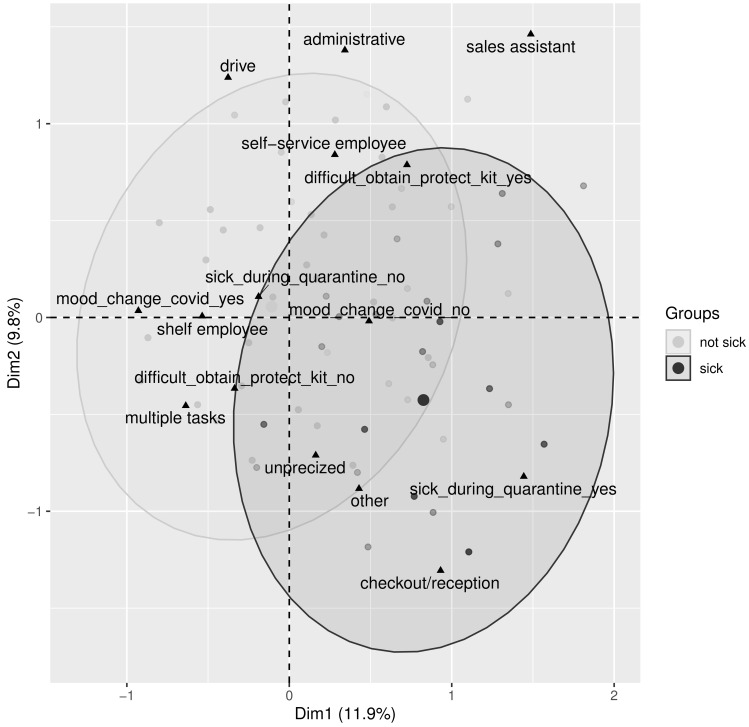
Multiple correspondence analysis (n=1519). The two first factorial dimensions are presented (a proportion of total explained variance is indicated for each axis); groups of sick and non-sick employees during the lockdown are indicated by the 95% confidence ellipses. The considered variables are: being or not being sick, having or not having difficulty to obtain a protection kit, position of the employee, and changed or not changed mood during lockdown.

**Figure 3 healthcare-10-01404-f003:**
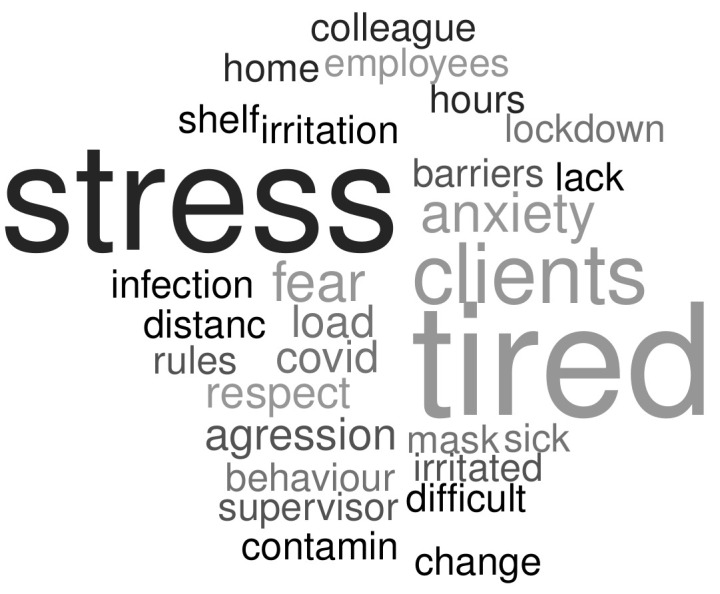
Mood change description of employees (derived from free answer).

**Figure 4 healthcare-10-01404-f004:**
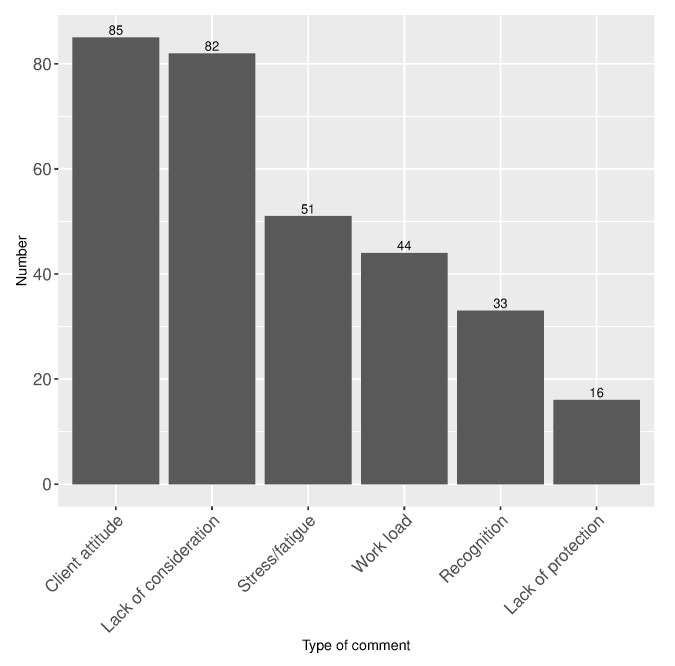
Analysis of free comments by employees (n=311 available comments).

**Figure 5 healthcare-10-01404-f005:**
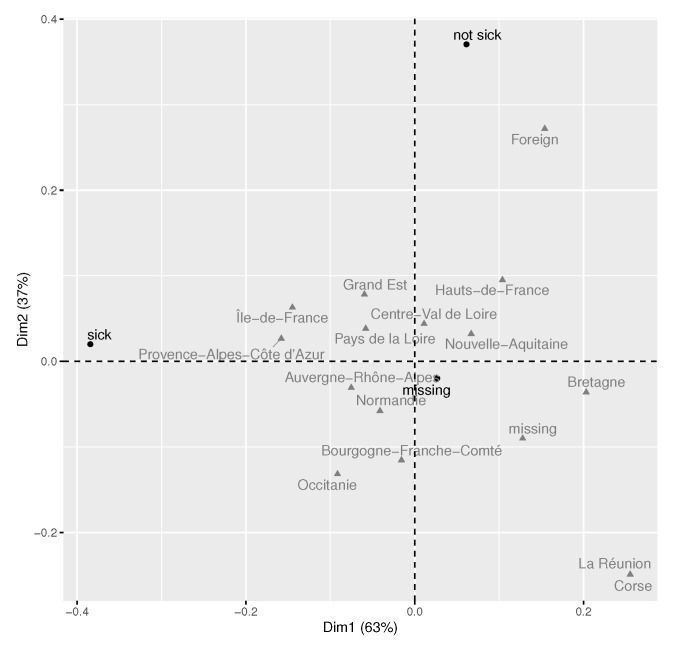
COVID by geographical region: correspondence analysis, employees (n=1746), the two first factorial dimensions are presented. A proportion of total explained variance is indicated for each axis.

**Figure 6 healthcare-10-01404-f006:**
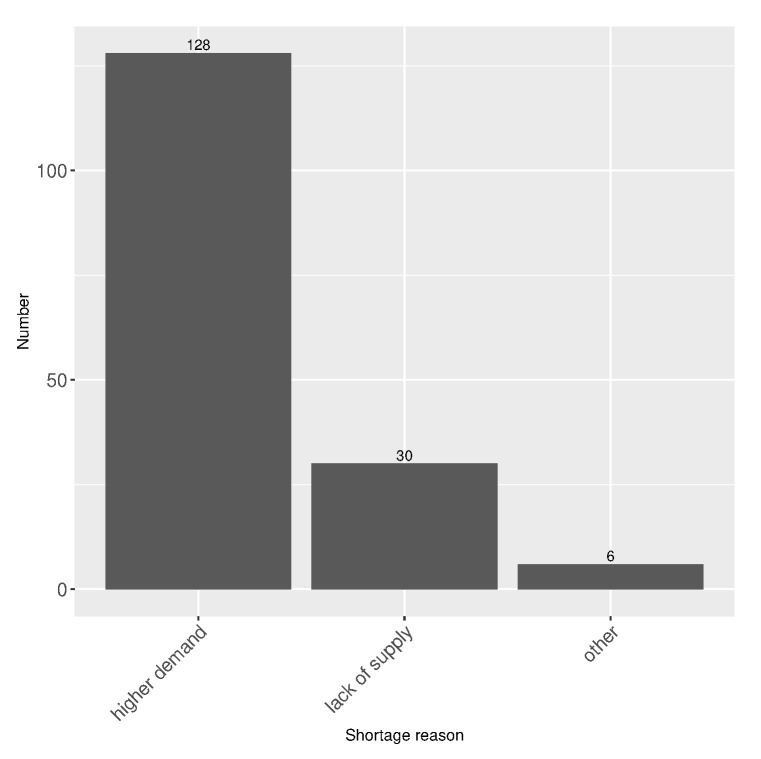
Shortage description reasons reported by supervisors (n=164, 5 missing answers, 2 responses not reporting shortage).

**Figure 7 healthcare-10-01404-f007:**
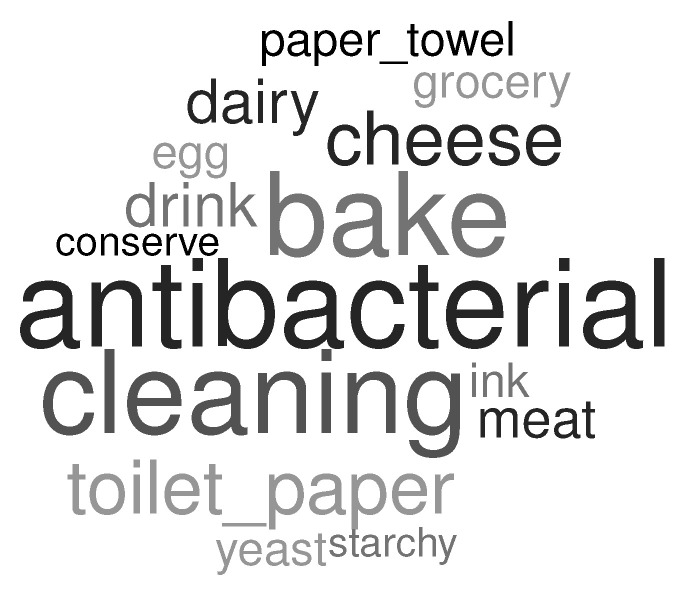
Detailed shortage description by supervisors (derived from free answers).

**Figure 8 healthcare-10-01404-f008:**
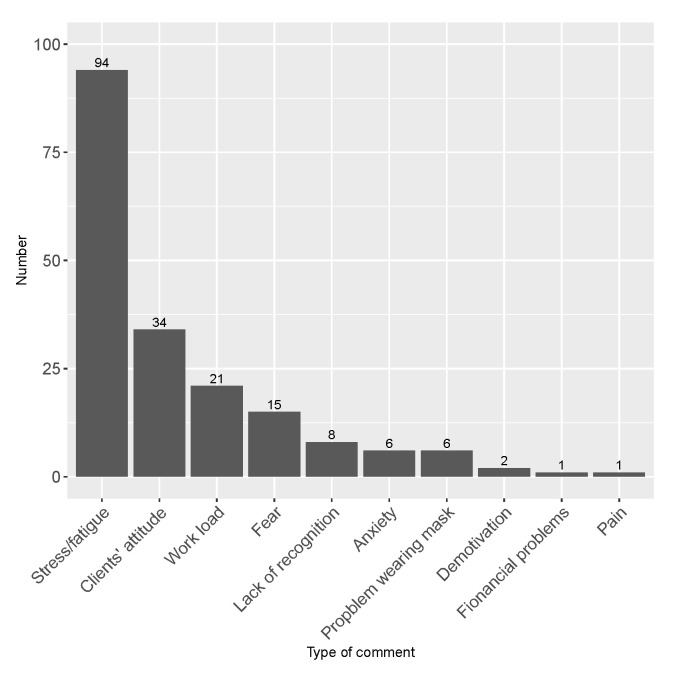
Mood change description by supervisors (n=188).

**Table 1 healthcare-10-01404-t001:** Reasons of not working during lockdown (*n* = 60).

Reported Reason	*n* (%)
Child care	23 (38.3%)
Sick leave	23 (38.3%)
Other	13 (21.7%)
Missing	1 (1.70%)

**Table 2 healthcare-10-01404-t002:** Descriptive statistics: employees working and sick during quarantine (n=194).

Characteristics	n(%)
gender:	
female	146 (75.3%)
male	48 (24.7%)
employed since	
<6 months	6 (3.16%)
6 months–1 year	15 (7.89%)
1–5 years	69 (36.3%)
5–10 years	34 (17.9%)
>10 years	66 (34.7%)
size of supermarket:	
grocery store	9 (4.71%)
supermarket	99 (51.8%)
hypermarket	83 (43.5%)
uniform change during lockdown	87 (44.8%)
difficulty to obtain protect kit	78 (40.6%)
mood change during lockdown	41 (21.1%)
work amount change	180 (93.8%)
position:	
administrative	15 (7.94%)
checkout/reception	38 (20.1%)
drive	8 (4.23%)
multiple tasks	8 (4.23%)
other	10 (5.29%)
sales assistant	6 (3.17%)
self-service employee	21 (11.1%)
shelf employee	55 (29.1%)
unprecised	28 (14.8%)
age:	
[18, 25]	33 (17.1%)
(25, 40]	112 (58.0%)
(40, 62]	48 (24.9%)
duration of sick leave, N (%):	
<1 week	99 (51.0%)
1–2 weeks	36 (18.6%)
2–3 weeks	23 (11.9%)
3–4 weeks	20 (10.3%)
>1 month	16 (8.25%)

**Table 3 healthcare-10-01404-t003:** Description of employees’ working status and COVID status (n=1746).

	COVID-Free	COVID-Confirmed	Missing Data on COVID Status
sick leave	8% (5)	10% (6)	82% (49)
work with COVID	4% (72)	7% (113)	89% (1486)
missing answer		7% (1)	93% (14)

**Table 4 healthcare-10-01404-t004:** Description of employees’ conditions reported by supervisors (n=165, the responses with irrelevant information concerning the number of employees were removed from analysis).

Characteristics	
proportion of full-time workers, mean (SD)	0.79 (0.18)
proportion of part-time workers, mean (SD)	0.20 (0.25)
proportion of absent workers, mean (SD)	0.12 (0.23)
proportion of workers with COVID, mean (SD)	0.04 (0.07)
proportion of absent due to child-care, mean (SD)	0.06 (0.10)
uniform change during lockdown, *n* (%):	
no	64 (38.8%)
yes	97 (58.8%)
missing answer	4 (2.42%)
initiative to change uniform, *n* (%):	
employees	13 (7.88%)
myself	21 (12.7%)
store	62 (37.6%)
missing answer	69 (41.8%)
difficulty to obtain protection kit, *n* (%):	
no	88 (53.3%)
yes	72 (43.6%)
missing answer	5 (3.03%)

**Table 5 healthcare-10-01404-t005:** Description of orders and turnover (normal time vs. lockdown). All chi-squared *p*-values comparing the two groups are <0.05. The percentage is calculated over n=171 supervisors for the number of visits and the turnover and over those concerned by e-commerce: n=88 at normal time and n=85 during the lockdown (see Appendix A (Table A3) for details).

	Normal Time	Lockdown
Number of visits		
[250, 2000]	23 (13.5%)	28 (16.3%)
(2000, 5000]	43 (25.1%)	29 (17.0%)
(5000, 10,000]	32 (18.7%)	32 (18.7%)
(10,000, 120,000]	28 (16.4%)	21 (12.3%)
missing answer	45 (26.3%)	61 (35.7%)
Number of e-commerce orders		
[0, 1)	1 (1.1%)	0 (0.0%)
[1, 30)	16 (18.2%)	5 (5.9%)
[30, 200)	30 (34.1%)	24 (28.2%)
[200, 3000]	21 (23.9%)	42 (49.4%)
missing answer	20 (22.7%)	14 (16.5%)
Turnover (euros)		
<20,000	18 (10.5%)	6 (3.5%)
20,000–50,000	25 (14.6%)	18 (10.5%)
50,000–100,000	29 (17.0%)	34 (19.9%)
100,000–150,000	20 (11.7%)	19 (11.2%)
150,000–200,000	17 (9.9%)	18 (10.5%)
200,000–500,000	33 (19.3%)	31 (18.1%)
>500,000	16 (9.4%)	17 (9.9%)
missing answer	13 (7.6%)	28 (16.4%)
E-commerce turnover (euros)		
<20,000	34 (38.6%)	17 (20.1%)
20,000–50,000	16 (18.2%)	25 (29.4%)
50,000–100,000	5 (5.7%)	11 (12.9%)
>100,000	11 (12.5%)	16 (18.8%)
missing answer	22 (25.0%)	16 (18.8%)

## Data Availability

All files are available from the Google Forms database: https://docs.google.com/spreadsheets/d/1EV8ux-_aqAo0ZF5-BeVd3ukbX5Aq8gHiDdA4GVxqMw0/edit?usp=sharing for the supervisors. https://docs.google.com/spreadsheets/d/1VjHWB2UwGsYXWp-q-nhL_b2aK7lWvq9fuOHpmbz_ouA/edit?usp=sharing for the employees.

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
