# Peer review of "The Impacts of COVID-19 Pandemic on the Food Sector and on Supermarket Employees in France during the First Lockdown Period"

_healthcare, 2022, doi:10.3390/healthcare10081404_

Round 1

Reviewer 1 Report

Thank you very much for inviting me to review the paper entitled “The impacts of COVID-19 pandemic on the food sector and on supermarket employees in France during the first lockdown period”. This research tries to know the different impacts of the pandemic on supermarket staff and on the food sector: COVID transmission among supermarket workers, working conditions, food supply, etc, during the first lockdown period due to the COVID-19 pandemic, from March 17 to May  11, 2020. According to this study, a lack of government recognition, lacking diagnostic tests, and increased workload were among the major challenges which supermarket staff has been faced during the lockdown period. Overall, I found it an interesting research topic. I hope my comments help you to improve the paper and make it publishable. 

I would like to thank you the authors of the paper, for keeping the method and data collection transparent. Since, you have used social media for data collection, it will be much better, if you can explain clearly about how you reached the target how you differentiate between supermarket supervisors and stuff during data collection process? How you made sure, the person who replied your survey is one of these groups of people. 

If I am not wrong, your survey was in French, how you treated people with low knowledge of this language? Did you distribute your survey in English or other languages as well? I know many immigrants working in supermarkets.  

Do you have any archival data about the number of infected and dead supermarket supervisors and staf during data collection process? 

The reader of your paper may interest to know what really was going on in France during your data collection process. I recommend you add some useful information about the total number of infected and dead people in France at the beginning of your study (March 17, 2020), and at the end of your study (May 11, 2020). You may use the following website 

https://www.worldometers.info/coronavirus/country/france/

I hope, you find my comments helpful. 

Author Response

Dear reviewer,

Please find attached our detailed responses to your comments.

Best regards,

Cyrielle Dumont and Génia Babykina

Reviewer 2 Report

This is an interesting study on an important occupational group during the COVID-19 pandemic. However, it is unclear how this paper and its topic fit with the scope of Healthcare. In addition, the manuscript has some significant flaws that preclude its consideration for acceptance.

1. I recommend an English speaker/writer work with the authors, as the writing style severely limits the paper.

2. The study population is a convenience sample of members of a Facebook group. There are no details about how many members are in the group and whether this study can be generalizeable to the French supermarket worker population.

3. Sections of the manuscript appear out of order. Lines 130-139 belong in the methods, and there are conclusions in the discussion. Table 1 is unnecessary.

4. The conclusions are not well supported by the results. It is unclear why the authors state that the participants were not “strongly affected” by COVID.

5. The missing data on COVID status is a severe limitation.

Author Response

(The authors gave the same response as above.)

Reviewer 3 Report

The manuscript is a cross-sectional study among supermarket staff who engaged in work during the COVID-19 lockdown period in France.

It should be mentioned every part of the study from methods to results has been written clearly and results are discussed well at the end of the study.

There are a few points that need to be corrected as follows:

1-      In lines 30 to 43, instead of mentioning the number of references it is better to say in (authors name study) e.g. In [11] the authors approach... can be: {In Ying et al. report...} or {in one study...}, then the reference will be placed in the appropriate place in the sentence.

2-      The line 69 to 70 is not necessary, better be removed.

3-      The “Research questions”   is not necessary as a separate sub-title and can be removed which are mentioned in table 1.

4-    Figure A1 needs to be replaced with one better justification on the page.

Author Response

(The authors gave the same response as above.)

Reviewer 4 Report

The paper presents a study focused on exploring the different impacts of the pandemic on supermarket staff and on the food sector in France: COVID transmission among supermarket workers, working conditions, food supply, etc.

The paper approaches a topical issue and the manuscript is clear and presented in a well-structured manner.

The cited references are relevant and most of them are published in the last 5 years.

Specific comments:

1)    Fig.1, 4, 6, 8 – The values for categories represented graphically should be indicated on the corresponding bars.

2)    Row 335 – replace “Stated” with “States”.

3)    Fig.A1 – the figure may need to be rearranged, section b) and d) are not completely visible in the .pdf version of the manuscript

Author Response

(The authors gave the same response as above.)

Reviewer 5 Report

Thank you for the opportunity to review this study entitled “The impacts of COVID-19 pandemic on the food sector and on supermarket employees in France during the first lockdown period” (healthcare-1796195).

The study focused on the effect of the COVID-19 pandemic, by investigating the impact of the first lockdown on supermarket staff and on the food sector in France.

In my opinion, the research topic is relevant, and the study is interesting. Parallelly, there are some issues that need to be addressed before the paper will be suitable for publication.

1.     Abstract: the information about the sample should be deepened (Mean age and SD? Percentage of men and women?) to provide a clear picture of what will be presented in the paper.

2.     Introduction: In my opinion, it would be good to refer to trend or longitudinal studies, if any. Since the authors frame this study considering the impact that COVID-19 has on a psychological level, I suggest some research to propose a comprehensive framework in the introduction, which should be supplemented with further literature search by the authors:

- Hyland et al., 2021; doi: 10.1016/j.psychres.2021.113905.

- Gori & Topino, 2021; doi: 10.3390/ijerph18115651

- Wang et al., 2020; doi: 10.1016/j.bbi.2020.04.028

To find the suggested articles, the authors can use this source: https://www.doi.org/

3.     Please, rephrase this sentence to improve its readability “The aims of the surveys are to assess the general situation in terms of COVID transmission among supermarket workers in terms of their working conditions as well as to obtain a feedback from employees concerning their mental and physical state during the lockdown period.”.

4.     In my opinion, lines 69-70 should be deleted.

5.     In my opinion, the “Research questions” could be moved to the end of the introduction when the research objectives are being clarified.

6.     More information about the used measures should be added. Have standardized self-report scales been used?

Author Response

Dear reviewer,

Please find attached our detailed responses to your comments.

Best regards,

Cyrielle Dumont and Génia Babykina

This manuscript is a resubmission of an earlier submission. The following is a list of the peer review reports and author responses from that submission.

Round 1

Reviewer 1 Report

  1. The introduction section is not written clearly, it would have been great if the author can follow the sequence as the revised introduction should have the following flow: brief overview -> importance of topic/domain -> research problem -> research gap--> aims/objectives -> overview of remaining sections. In particular, this study should deal with the contribution to the research problem a little more in the introduction. The research gaps and theoretical contributions are not strong enough. The paper did not present a strong explanation in terms of the research gap and research questions.
  2. The authors need to expand the section on implications. They have some interesting findings but need to develop them more and put more emphasis on how practitioners can benefit.

Reviewer 2 Report

please find my comments in the attached material
